# Influencing the Crystalline Domains of Poly(vinylidenedifluoride) Composites Using Fluorinated Silica Nanoparticles as Drop-In Modifiers

**DOI:** 10.3390/molecules27238398

**Published:** 2022-12-01

**Authors:** Nathan J. Weeks, Cole R. Phelps, Enrique T. Gazmin, Scott T. Iacono

**Affiliations:** Laboratories for Advanced Materials, Chemistry Research Center, Department of Chemistry, United States Air Force Academy, Colorado Springs, CO 80840, USA

**Keywords:** fluoropolymers, polyvinylenedifluoride, nanocomposites, silica nanoparticles, monolayer

## Abstract

Improvements to fluoropolymer processing techniques by way of utilizing nanoparticles as drop-in processing aids have pronounced effects on bulk composite properties. In this work, we prepared fluoroalkyl-silanized silica nanoparticles (F-SiNPs, ca. 200 nm) that were solvent-blended with polyvinylenedifluoride (PVDF) in order to prepare composites with varying weight fractions. We demonstrated that the ability to functionalize SiNPs with long fluoroalkylchains that induced co-crystallization with the PVDF matrix, resulting in uniform particle dispersion and improved interlaminate adhesion. This was quantitatively investigated using calorimetry and thermogravimetric analysis, which showed a decrease in the bulk crystallinity of the virgin PVDF from 37% to 10% with minimal 10 wt % F-SiNP loading, rendering a nearly amorphous PVDF. Additional discussions in this work include the effects of various bare and fluoroalkyl-functionalized SiNP loadings on the amorphous and crystalline domains of the PVDF matrix, as well as thermal decomposition.

## 1. Introduction

Fluoropolymers continue to grow in the global economy and have been extensively used in high performance commercial applications, including, but not limited to, waterproof and corrosion resistant coatings, high-pressure tubing, and cryogenic service gaskets, and these topics have been recently reviewed [1]. It may come to the reader’s attention that fluoropolymers have recently drawn attention given their concern for human and environmental health and therefore several conflicting debates on the regulation of fluoropolymers have been thoroughly investigated [2,3]. Nonetheless, as it currently stands as a commercially, globally used commodity, many of the desirable properties of chain-growth fluoropolymers stems from their inherent high degree of crystallinity due to the unique arrangement of the fluorocarbon backbone deriving from the respective vinyl monomer feed stocks. In one extreme, fluorocarbon-saturated poly(tetrafluoroethylene) -*CF*_2_*CF*_2_- (PTFE or trade-named Teflon), for example, is estimated to have up to 90% crystalline domains, which prohibits solvent and melt processing of PTFE, whereby only energy-intensive machining or heterogeneous dispersions are required to achieve desired components, which leads to sacrificial materials or solvent waste as a consequence [4]. On the other hand, partially fluorinated poly(vinylenedifluoride) -*CF*_2_*CH*_2_- (PVDF) is one of the most versatile fluoropolymers used in a plethora of manufacturing techniques and is highly adaptable to solvent blending for films and coatings, as well as melt processing for injection molded parts or drawn fibers [5]. Additionally, the ability to incorporate fillers such as particles and fibers, for example, demonstrates PVDF’s greater utility by manipulating its crystalline microstructure to enrich desired β-phase domains in order to hone in on its intrinsic properties as a novel piezoelectric material [6].

The unrealized potential of bulk physical property manipulation by incorporating heterogeneous fillers into fluoropolymers by way of melt processing is often limited. This is because of an inherent low surface energy due to high fluorine content that causes particle or fiber delamination, as well as elevated temperature processing due to a high crystalline content. This study focuses on PVDF as an ideal candidate matrix to serve as a host for modified fillers that might reduce the crystalline domains while retaining the desired hydrophobic properties. Specifically, the use of fluoroalkyl-modified silica nanoparticles (F-SiNPs) with an average diameter of 200 nm were prepared from a facile chemical silanization of silica nanoparticles (SiNPs) [7], which have been shown to significantly enhance the bulk properties of various hybrid composites in a recent review [8]. While the application of processing unmodified, mesoporous SiNPs into PVDF has been prepared most recently as a nucleation agent to improve surface energy and piezoelectric response [9,10], and reviewed extensively [11], only a single report has demonstrated SiNPs decorated with grafted PVDF [12]. F-SiNPs, as well as discreetly sized molecular silicas have been used extensively for notably enhancing the surface energy properties of primarily hydrocarbon-based polymers [8,13] and limited to one fluoropolymer example with Viton, a terpolymer of poly(ethylene), PTFE, and perfluoromethylvinyl ether [14]. However, to our knowledge, there are no reports of F-SiNP- and PVDF-blended compositions. In order to address this research gap, this work seeks to demonstrate the facile preparation of low crystalline, solvent-blended SiNP/PVDF composites whereby one can either: (1) obtain nearly amorphous PVDF composites for useful optical applications and/or (2) formulate a precursory raw material that can then be utilized for complex melt-processing techniques. This was simply achieved by using mesoporous F-SiNPs as a drop-in modifier for processing PVDF, as illustrated in Figure 1.

## 2. Results and Discussion

### 2.1. Preparation of Silica Nanoparticles (SiNPs) and Thier Surface Functionalization to Fluorinated Silica Nanoparticles (F-SiNPs)

The chemical modification of SiNPs is illustrated in Figure 1. Bare SiNPs, typically in 20 g batches, were prepared from the common base-catalyzed Stöber process in order to prepare a uniform distribution of 200 nm (avg size) diameter particles, as confirmed by AFM and SEM analysis. The chemical modification of SiNPs with tridecafluoro-1,1,2,2-tetrahydrooctyl)dimethylchlorosilane afforded F-SiNPs, whereby TGA analysis produced a 15 wt % loss of organic material at 900 °C in argon, which compares with previous work [7] that a uniform fluoroalkane monolayer was successfully achieved.

### 2.2. Formulation and Thermal Analysis of SiNPs and F-SiNPs Solvent-Blended PVDF Composites

PVDF was successfully solvent-blended with SiNPs and F-SiNP up to a 10 wt % loading of F-SiNPs, affording white, opaque, flexible, free-standing films. The summary of thermal analysis by DSC and TGA is shown in Table 1. Loadings higher than 10 wt % F-SiNPs were attempted and lead to visually significant phase separation, rendering the resulting films brittle and were not studied for thermal analysis, given that the lack of utility and inhomogeneity of the specimens would render analysis inconsistent.

DSC analysis revealed two discrete melting temperature transitions (*T*_m_) of 167–168 °C and 171–172 °C from SiNP and F-SiNP composites, regardless of loading, compared with virgin solvent-processed PVDF. For visualization of the heating and cooling events, Figure 2 provides various F-SiNP loadings in PVDF. As demonstrated, the enthalpy of the melting transition (Δ*H*_m_) decreased from 30.4 J/g to 16.7 J/g and 44.2 J/g to 10.6 J/g with increased loadings of SiNPs and F-SiNPs, respectively. Significant decrease in Δ*H*_m_ was observed at 10 wt % loadings of both SiNPs and F-SiNPs, indicating that the particles contributed to the alteration of the crystalline domains upon solvent processing. This was most affected by the F-SiNPs as a result of the better interdigitating of the fluoroalkyl chains with the amorphous domains of the PVDF polymer chains, which has also been observed similarly with other metal oxide modifiers in PVDF [15]. The modest increase in the Δ*H*_m_ of the 1wt % F-SiNP/PVDF (44.2 J/g) composite compared with the virgin-processed PVDF (39.0 J/g) suggests, at low loading, the F-SiNPs may induce the nucleation of PVDF crystalline domain formation. Surprisingly, given the effects of the SiNPs and F-SiNPs on Δ*H*_m_, ATR-FTIR analysis on all solvent-processed/blended films showed no alteration in distribution of PVDF microstructure of the α, β, γ-crystalline phases.

The cooling behavior, as illustrated in Figure 2, of the melted composites was measured by observing the crystallization temperature transition (*T*_c_), revealing no significant deviation of 141–143 °C from the various wt % SiNPs- and F-SiNPs-blended composites and was only slightly suppressed compared with the virgin solvent-processed PVDF *T*_c_ at 139 °C. As expected, the heat of crystallization (Δ*H*_c_) parallels with the Δ*H*_m_ trend for SiNP/PVDF-blended composites by decreasing linearly with increasing 1–10 wt % from 44.7 J/g to 22.5 J/g. However, the F-SiNP/PVDF solvent-processed films showed a decrease starting only at 5 wt % (45.9 J/g) and a significant decrease at 10 wt % (19.4 J/g) compared with the virgin PVDF (50.1 J/g). This indicates that at high F-SiNP loading, the fluoroalkyl-modified SiNPs retard the crystallization of the PVDF domain formation, while at low loadings (below 5 wt %) the fluoroalkyl chains remain miscible. As such, the percent crystallinity (*X*_c_), determined from Δ*H*_m_ of 100% crystalline PVDF (104.5 J/g), drops linearly with increasing bare SiNPs, but F-SiNPs show steady *X*_c_ until 10 wt %, where we nearly eliminate PVDF crystalline domain formation (10%). Lastly, the glass transition temperature *T*_g_ was observed at −37 °C for the virgin solvent-processed PVDF, as well as the SiNP- and F-SiNP-loaded composites, indicating that the presence of the 200 nm-sized SiO_2_ particles has no influence on amorphous PVDF domains.

The thermal performance of the composites was evaluated by TGA in argon by measuring the sharp step-wise onset decomposition temperature (*T*_d_) and resulting char yield at 900 °C. PVDF composites loaded with bare SiNPs dramatically accelerated the degradation of PVDF when heated. For example, at a 10 wt % loading of SiNPs, a decrease of nearly 100 °C *T*_d_ onset compared with virgin PVDF indicates the presence of available free silanols (Si-OH) on the SiO_2_ particle surface, which induces expected acid-catalyzed decomposition of PVDF by HF elimination. Contrary, all PVDF composites blended with various F-SiNP loading reduce the presence of free silanols and maintain the thermal stability of the virgin. PVDF matrix observed with onset *T*_d_ of 425 °C in Ar. The resulting char yields of the F-SiNP/PVDF-blended composites were observed to be statistically consistent with the amount NP loading resulting in residual SiO_2_ and carbonaceous char.

## 3. Materials and Methods

### 3.1. Materials

All reagents, unless otherwise noted, were obtained from commercial sources and used as received. Poly(vinylenedifluoride) (PVDF) high-flow homopolymer grade Kynar 705 was generously purchased from Arkema (King of Prussia, PA, USA). (1H,1H,2H,2H-perfluorooctyl)dimethylchlorosilane (PFOMCS) was purchased from Gelest (Morrisville, PA, USA).

### 3.2. Instrumental Methods

Attenuated total reflectance Fourier transform infrared (ATR-FTIR) spectra were collected using a Thermo Nicolet FTIR spectrometer iS10 (Waltham, MA, USA). Differential scanning calorimetry (DSC) was performed on a TA Instruments (New Castle, DE, USA) Auto Q20 instrument in nitrogen. Samples (ca. 5 mg) were sealed in aluminum hermetic pan with an empty sealed hermetic pan serving as the reference. Thermal transitions were reported on the third heating cycle and values were recorded at the mid-point unless otherwise specified. Samples were heated/cooled at a rate of 5 °C/min. Thermal gravimetric analysis (TGA) was performed on a TA Instruments (New Castle, DE, USA) Q500 instrument at a scan rate of 5 °C/min in argon. Samples (5–10 mg) were measured with a platinum crucible and heated from room temperature to 900 °C. TA Universal Analysis 2000 graphical software was used to determine the thermal properties. Scanning electron analysis (SEM) was captured with a Tescan VEGA (Brno, Czechia) equipped with elemental analysis. Atomic force microscopy measurements were performed on Park Systems (Suwon, Republic of Korea), employing Park SmartAnalysis™ graphical software suite to determine particle size measurements.

### 3.3. General Procedure for the Prepartion of Silica Nanoparticles (SiNPs) and Fluorinated Silica Nanoparticles (F-SiNPs)

Silica nanoparticles (SiNPs, avg 200 nm by SEM or AFM) were prepared by the method of Stöber [16]. Fluorinated silica nanoparticles (F-SiNPs, avg 200 nm by SEM) were prepared by using a published procedure [7] using tridecafluoro-1,1,2,2-tetrahydrooctyl)dimethylchlorosilane.

### 3.4. General Procedure for Preparation of Solvent-Blended SiNP PVDF Nanocomposites

Nanocomposites were prepared by solvent blending a stock solution of PVDF dissolved in a minimal amount of reagent grade *N*,*N*-dimethylformamide (DMF) (typically, 10 wt % PVDF loading) with either SiNPs or F-SiNPs to the desired weight percent, mechanically mixed for 1 h, then drop cast onto a Petri dish, allowed to evaporate the DMF for 16 h in a fume hood, and finally placed in a vacuum over for 16 h at 60 °C to facilitate complete drying prior to analysis.

## 4. Conclusions

In summary, we reported the preparation of bare and fluoroalkyl surface-modified SiNPs that were solvent-blended with PVDF affording composites with modified bulk crystalline properties. While the bare SiNPs provided a suitable comparison to the influence of melting and crystallization temperatures, they offered limited utility due to accelerated degradation of PVDF under demanding thermal processing or cycling conditions. The F-SiNPs, on the other hand, offered a viable path forward serving as a melt-processing aid for these PVDF composites, while retaining thermal stability to that of the virgin PVDF. Using F-SiNPs with modest loadings up to 10 wt % nearly eliminated the crystalline domains of PVDF, offering an entirely amorphous matrix for potential optical applications. Furthermore, the incorporation of F-SiNP solvent blended with PVDF afforded composites that showed no plasticization effects nor reinforcing effect, which is ideal for controlling processing parameters for bulk components such as injection molding, as well as additive manufacturing applications. Continued work on piezo-optical responses, rheological aspects, and mechanical testing on manufactured parts will be the focus in on-going investigations.

## Data Availability

Not applicable.

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
