# Peer review of "Influencing the Crystalline Domains of Poly(vinylidenedifluoride) Composites Using Fluorinated Silica Nanoparticles as Drop-In Modifiers"

_molecules, 2022, doi:10.3390/molecules27238398_

Round 1

Reviewer 1 Report

Authors present interesting results on the effect of silica silanization with perfluoroalkyl chains on the crystallinity and thermal properties of Silica/PVDF composites. I would suggest to include, at least as supplementary information: 1) calorimetric traces to show the two transitions, 2) FTIR spectra of two relevant compositions showing the effect of a decrease in crystallinity

Reviewer 2 Report

1.There is a lack of charts or figures of structural characterization and thermochemical analysis in this work, such as the figures of SEM and AFM mentioned in 3.1 and charts of DSC mentioned in 3.2. Readers may have a more visualized impresson on what you want to show them by the exposition and data given in these sections if you replenish them.

2.Please explain the mechanism why modified fillers, such as F-SiNP, could reduce the crystalline domains of PVDF in introduction more concretely with more references and examples.

3.Since crystallinity of PVDF decreases little at low F-SiNP loading, it might be better to show which loading of F-SiNP result in significant influence on crystalline domains of PVDF.

4. The article have discussed the influence that SiNP and F-SiNP cause in the aspects of decreasing crystalline domains and thermostability. However, it is also needed to indicate that whether there would be effects on other properties, such as mechanical property or elasticity, after dropping in SiNP and F-SiNP.

5.You mentioned preparing PVDF composites for optics application, but didn’t discuss optic properties in your research. It would be better to add some characterizations of optic properties.

Author Response

See attachement

Reviewer 3 Report

The authors successfully improved improvements to fluoropolymer processing techniques by way of utilizing nanoparticles as drop-in processing aids has pronounced effects on bulk composite properties. A few points can make the manuscript easier to understand for readers.

Point 1: How to prove that silica nanoparticles (SiNPs) have chemically grafted with tridecafluoro-1,1,2,2-tetrahydrooctyl)dimethylchlorosilane, I have not seen relevant data proof in the article, please explain it.

Point 2: The authors mentioned that “Loadings higher than 10 wt% F-SiNPs were attempted and lead to visually significant phase separation rendering the resulting films brittle”. How to prove that it has obvious phase separation, whether there is relevant experimental data to prove it, if so, please add.

Point 3: The authors mentioned that “This was most affected by the F-SiNPs as a result of  better interdigitating of the fluoroalkyl chains with the amorphous domains of the PVDF Molecules polymer chains”. Corresponding references should be added.

Point 4: The Char yields of SiNP/PVDF blend composites was not consistent with the amount SiNP, please explain it. 
